# Emotional expression through musical cues: A comparison of production and perception approaches

**Annaliese Micallef Grimaud**⊙*, **Tuomas Eerola**⊙

Department of Music, Music and Science Lab, Durham University, Durham, United Kingdom

* annaliese.micallef-grimaud@durham.ac.uk

**Data Availability Statement:** All data and stimuli used are available on OSF repository: https://osf.io/atxhk/.

**Funding:** AMG is funded by the Tertiary Education Scholarships Scheme (TESS) 2019 grant, financed

## Abstract

Multiple approaches have been used to investigate how musical cues are used to shape different emotions in music. The most prominent approach is a perception study, where musical stimuli varying in cue levels are assessed by participants in terms of their conveyed emotion. However, this approach limits the number of cues and combinations simultaneously investigated, since each variation produces another musical piece to be evaluated. Another less used approach is a production approach, where participants use cues to change the emotion conveyed in music, allowing participants to explore a larger number of cue combinations than the former approach. These approaches provide different levels of accuracy and economy for identifying how cues are used to convey different emotions in music. However, do these approaches provide converging results? This paper's aims are two-fold. The role of seven musical cues (tempo, pitch, dynamics, brightness, articulation, mode, and instrumentation) in communicating seven emotions (sadness, joy, calmness, anger, fear, power, and surprise) in music is investigated. Additionally, this paper explores whether the two approaches will yield similar findings on how the cues are used to shape different emotions in music. The first experiment utilises a production approach where participants adjust the cues in real-time to convey target emotions. The second experiment uses a perception approach where participants rate pre-rendered systematic variations of the stimuli for all emotions. Overall, the cues operated similarly in the majority (32/49) of cue-emotion combinations across both experiments, with the most variance produced by the dynamics and instrumentation cues. A comparison of the prediction accuracy rates of cue combinations representing the intended emotions found that prediction rates in Experiment 1 were higher than the ones obtained in Experiment 2, suggesting that a production approach may be a more efficient method to explore how cues are used to shape different emotions in music.

## Introduction

An important aspect of music is that it can communicate different emotional expressions to the listener [1, 2]. A substantial amount of previous literature suggests that composers and

by the Ministry for Education, Sport, Youth, Research and Innovation in Malta. URL: https://education.gov.mt/en/education/myScholarship/Pages/TESS—Tertiary-Education-Scholarship-Scheme.aspx The funders had no role in study design, data collection and analysis, decision to publish, or preparation of the manuscript.

**Competing interests:** The authors have declared that no competing interests exist.

performers can successfully encode a specific emotional expression in the music using particular musical cues (i.e., properties of the music) to communicate it to the listeners. In turn, listeners use these same cues to decode the emotion communicated and, in general, can successfully identify the intended emotion conveyed [3–5]. The musical cues tempo, mode, pitch level, dynamics, timbre, rhythm, melodic range and direction, and harmony have all been identified as having an influence on the emotional expression shaped in the music (for an overview, see Juslin & Lindström, 2010). Understanding how musical cues affect the different emotions communicated through real music to the listeners has important applications, such as investigating emotion development and regulation in children and teenagers [6, 7], utilising music as a medium for non-verbal patients [8], and encapsulating specific branding identities in music for marketing purposes [9], to name a few.

Although musical cues and emotional expression have been investigated over the last century, research has only scratched the surface of how musical cues operate and shape the different emotion profiles. A number of studies have suggested that it is the additivity of musical cues that helps convey different emotions in the music, rather than the effect of an individual cue [2, 10–12]. However, the role of multiple musical cues as a combination has not been investigated as much [4, 13, 14]. Previous research tended to focus on one musical cue, such as mode [15, 16], timbre [17–21], melody [14], harmony [22], or harmonic intervals [23]. Other studies investigated two to three cues simultaneously, each with a limited number of variations/levels (e.g., tempo fast/slow) [14, 24–27], and only a few studies have tried to explore a bigger cue space with seven or eight cues and multiple cue levels simultaneously and their interactions [4, 11].

The most prominent methodology used to investigate how musical cues affect the emotional expression communicated through music is a perception approach, where similar musical excerpts are systematically created by slightly varying the levels of different cues. These musical variations would then be assessed by participants evaluating the excerpts in terms of emotional expression [4, 10, 11, 15, 24, 28]. This approach allows for minute changes in musical cues to be investigated, with complete experimental control. However, each systematic variation produces another musical stimulus that participants would need to listen to and evaluate. Therefore, the number of cues and cue level combinations that can be investigated simultaneously utilising a systematic manipulation design and perception study is limited, as a design with a large number of cue combinations becomes quickly unfeasible [29]. Furthermore, running numerous systematic variations on a musical stimulus might tamper with the ecological validity of the music [30].

An alternative method used in musical cues and emotion research is the production approach, where participants are in charge of changing a selection of musical cues in real-time to express different emotions through music. This methodology is referred to as analysis-by-synthesis [31], and this interactive paradigm allows for a larger cue space to be explored, as cue levels and combinations do not need to be pre-defined and rendered. Only a few studies have employed this methodology, with participants using either physical or digital sliders [6, 7, 32–34], or a one-key apparatus [35, 36] to express three to five emotions by controlling three to seven cues. A downside to these studies was that cues only controlled either the melodic part of the musical stimuli or Bach chorales chord sequences, which perhaps are not the best representatives of real music.

Other researchers have used correlation studies to assess which cues help communicate certain emotions in music. This is usually attained by first asking composers to create music expressing different emotions or using already existing music from a repertoire, then asking listeners to assess which emotion or valence/arousal state is being portrayed by the music and finally, analysing the score to identify which cue combinations correlate to different emotions

[37–39]. However, this methodology does not allow for the dissociation of the cues used. Thus, findings can only describe the effect of specific cue combinations, which cannot be freely changed and do not tell us the causal effect of the individual cues.

The use of different methodologies and their specific limitations begs the question of which methodology should be used to explore better the large cue space that exists and the cue combinations that help shape different emotions in real music. Furthermore, the use of different methodologies raises the questions of how reliable the methodologies are, and whether we are controlling for the potential involuntary effect of the chosen approach on the results, which may lead to a divergence in results.

This paper aims to investigate a combination of seven musical cues in relation to seven emotional expressions with the intent of exploring the rich and complex cue space that underlies expression in music. Furthermore, this paper aims to present a critical evaluation of two different approaches which are used in music and emotion research by carrying out the investigation of the musical cues and emotional expressions in question across two studies. Firstly, a production experiment will be carried out (Experiment 1), where participants will be in charge of shaping different emotional expressions in music by manipulating a selection of available cues via a computer interface. This approach will allow the participants to navigate through numerous cue combination possibilities in order to explore a substantial area of the cue space at once. Secondly, a perception experiment will be carried out (Experiment 2), where a pre-defined number of cue combinations and levels of the same seven cues will be systematically manipulated to create musical variations stemming from the original stimuli used in Experiment 1, which will then be evaluated on their emotional content by participants. A comparison of results between the two experiments will then be carried out, allowing for an exploration of whether the production study will produce reliable findings, thus confirming the suitability and efficiency of the approach, whilst also assessing the usability and efficiency of the traditional perception approach.

A combination of seven musical cues will be investigated in the present study: tempo, pitch, dynamics, brightness, articulation, mode, and instrumentation. Previous literature suggests that tempo, mode, and dynamics are three of the strongest contributing factors in emotional expression in music [25, 40, 41]. Pitch, articulation, brightness, and instrumentation have also been linked to affecting emotion perception in music [4, 7, 11, 20, 21, 39]; however, they have not been studied as much as the former cues. Therefore, this paper aims to explore how these cues and their combinations are used in shaping seven different emotional expressions in music. The emotional expressions investigated in this paper are sadness, joy, calmness, anger, fear, power, and surprise. Previous literature suggests that these seven emotions may be expressed through music [1, 2, 5, 42], with joy, sadness, anger, and fear being the most accurately recognised emotions by listeners, also at a cross-cultural level [26, 35, 43, 44]. Furthermore, these emotions cover a broad range on the emotion spectrum [45] and the valence-arousal circumplex model [46].

In summary, our research questions are:

(1) How do the musical cues and their combinations contribute to the expression of different emotions in music?

(2) To what extent do the results from the two experiments converge?

The first section of this paper details the production experiment (Experiment 1). The second section reports the perception experiment (Experiment 2). The following section compares the findings of Experiment 1 and 2, highlighting similarities and differences between the two experiments and the existing literature. Finally, the last section outlines the pros and cons of the two approaches utilised and gives insight on methodological considerations for future studies.

## Experiment 1: Production approach

In the production experiment, participants actively engaged with a computer interface called *EmoteControl* [47], which allows users to alter instrumental musical pieces via tempo, articulation, pitch, dynamics, brightness, mode, and instrumentation cues. This paradigm allows us to gain a deeper insight into the tempo, pitch, dynamics, and brightness cues as they are not confined to pre-determined distinct cue levels, whilst also exploring different levels of articulation, mode, timbre, and the cue combinations created by these seven cues.

### Method

**Participants.** Participants were recruited via social media and word-of-mouth. Forty-two participants (19 men, 23 women) between the ages of 20 and 68 years (*M* = 34.45, *SD* = 13.64) took part in the study. A one-question version of the Ollen Music Sophistication Index (OMSI) [48, 49] was utilised to distinguish between the participants' levels of musical expertise. Eight of the participants were musicians, whilst the remaining 34 were non-musicians. Participation in the study was voluntary.

**Material.** Seven tonal instrumental musical pieces ranging from 15 to 32 seconds in duration were utilised as material. These pieces were derived from a previously validated musical stimulus set specifically composed by Micallef Grimaud and Eerola [34] to be utilised with the *EmoteControl* interface. The seven musical excerpts had been validated via an online listening study as conveying one of the following seven emotions: joy, sadness, calmness, power, anger, fear, or surprise by having participants rate on Likert scales how much of each emotion was portrayed in the excerpts [34]. In this current study, participants were presented with these seven musical excerpts and asked to convey each of the seven emotions attributed to the stimulus set (joy, sadness, calmness, power, anger, fear, and surprise) in all the seven excerpts.

**Apparatus.** A second version of the computer interface *EmoteControl* (V2.0) was utilised for this experiment [47], which allows users to alter seven cues (tempo, articulation, brightness, pitch, dynamics, mode, with the instrumentation cue being the new addition in this second version of the interface) of instrumental musical pieces in MIDI format. A representation of the interface can be seen in Fig 1.

The interface is aimed at a general population, and no prior musical skills are required to utilise the interface. For this reason, layperson terms are used to describe certain words that are music-specific, such as 'mode' and 'articulation'. As can be seen in Fig 1, Mode is referred to as 'Change Pitch Alphabet', with Alphabet 1 being Major mode and Alphabet 2 being Minor. Articulation is referred to as 'Playing Method' in the interface, with detached indicating *staccato* and smooth referring to *legato*.

Cue changes for tempo, pitch, dynamics, and brightness are made via digital sliders in the interface. Digital buttons are used to switch between discrete levels of mode (pitch alphabet), articulation (playing method), and instrumentation. Changes to the music through the cues are instantly heard in real-time. When a MIDI file is inputted in *EmoteControl*, the properties of the musical piece are altered depending on the initial values of the cue sliders. Therefore, the users would not be exposed to the original version of the piece as it initially portrayed its intended emotion. The cue values were recorded at 10Hz.

### Cue details

**Tempo.** The tempo cue is measured in beats per minute (bpm). The slider is set with a minimum value of 40 bpm and a maximum value of 210 bpm to cover a wide tempo range.

**Articulation.** The articulation cue denoted as 'playing method' gives the option between *legato* (smooth) and *staccato* (detached).

**Pitch.** The pitch slider controls a pitch shift range of ±2 semitones from the starting point.

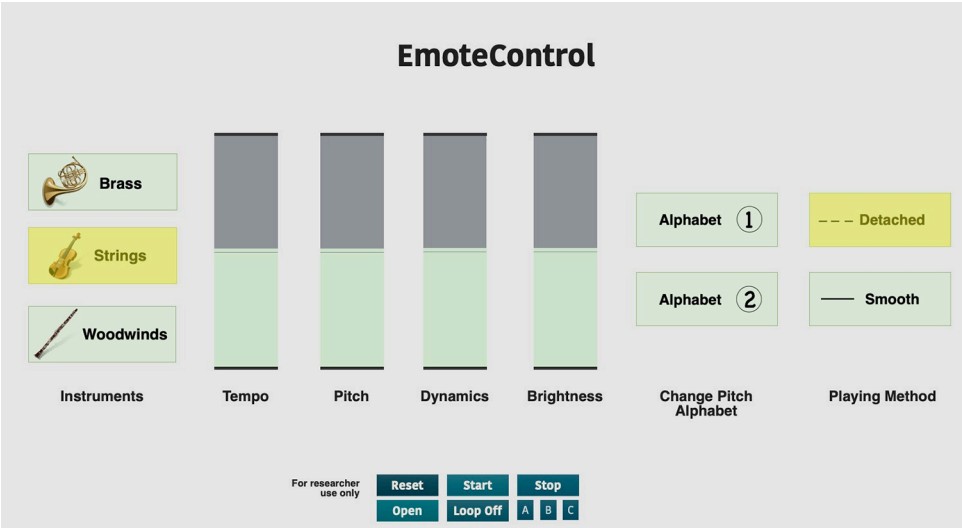

**Fig 1. The *EmoteControl* V2.0 user interface.**

**Dynamics.** The dynamics slider alters the MIDI volume of the virtual instrument used as sound output, rather than the overall volume via the dB level. The dynamics slider has a minimum MIDI volume value of 30 and a maximum value of 129.

**Brightness.** The brightness cue changes how bright or dull the musical piece sounds by altering the number of harmonics present in the sound. This is attained by changing the cut-off frequency value of a low-pass filter, with an available cut-off range of 305 Hz to 20,000 Hz. The low-pass filter has a steep slope gradient of 48dB/Oct and a Q factor of 0.43 to diminish frequency resonance.

**Mode.** Mode (labelled as pitch alphabet) gives participants the option to select a major mode denoted as pitch alphabet 1 or a harmonic minor mode (flattening the third and sixth degree of the scale to switch from major to minor) denoted as pitch alphabet 2.

**Instrumentation.** Participants can also choose which group of instruments play the music: brass, strings, or woodwinds. Previous findings have suggested that difference in sound attributes such as brightness and spectral entropy may impact the emotional quality of the music [11, 32]. Different instruments have been investigated with respect to emotional qualities, however, mostly as individual instruments rather than ensembles [50–52]. We wanted to test groups of instruments rather than individual instruments as this allowed us to test polyphonic music and a bigger register range simultaneously. Based on a pilot experiment that investigated the emotional expressivity range of a number of instruments (detailed in the S1 File) and using instruments with register ranges that could support the pitch ranges of the musical stimuli, the following instruments were chosen for the instrument family ensembles:

- Vienna horn (#3 in emotional expressivity range rank, S1 Table in the S1 File), piccolo trumpet, euphonium (#6), and trombone for the brass ensemble

- violin (#4), viola (#13), cello (#2), and double bass for the strings ensemble

- flute (#1), oboe (#8), clarinet (#9), and bassoon (#12) for the woodwinds ensemble

## Procedure

Full ethical consent was sought and approved by the Ethics Committee of Durham University before testing (*MUS-2020-06-08T10:17:29-cfsg56*). All participants were first given a detailed

description of the experiment and they provided written, informed consent. It is to be noted that this experiment was carried out during the COVID-19 pandemic; thus, specific safety measures were taken into consideration, detailed in Section 2 of the Supplementary Material. The first part of the experiment required participants to answer some demographic questions such as age, gender, and musical expertise. This was administered online via a short survey on Qualtrics. Instructions and a video demonstration for the second part of the experiment (the musical task using *EmoteControl* V2.0) were also presented to the participants online.

The musical task was done in person. Participants were presented with the *EmoteControl* V2.0 interface and instructed to use the seven cues available to change the music given to convey specific emotions designated by the researcher. Overall, all seven musical pieces were altered to convey the seven designated emotions. This yielded 49 different musical piece and emotion combinations. As fatigue might have set in if the same individual carried out 49 combinations, participants were split into three groups and given a subset of the total combinations. Each group carried out 14 unique piece and emotion combinations consisting of all seven musical pieces to convey two different emotions (7 musical pieces x 2 target emotions = 14 combinations). In addition, all groups carried out seven more combinations where participants had to portray the emotion already attributed to the different pieces (e.g., the piece composed and validated as conveying anger was altered by participants to express anger) to provide a common frame of reference, totalling 21 combinations (14 different trials per group + 7 trials common across groups = 21 combinations per group). Cue level value alterations made by participants for all seven cues were recorded for each trial. Prior to the musical task, participants were subjected to a practice trial where they changed the cue levels of a musical piece that was not utilised during the actual experiment to get accustomed to the interface and the musical task at hand. The experiment took approximately 30 minutes to complete.

## Results

The programming language R was used within the RStudio environment version 1.2.1335 to analyse the data collected. First, we examined the consistency of participant cue usage by calculating the inter-rater agreement within each block of 21 stimuli and emotion combinations across each cue and participant, using Cronbach's alpha (intraclass correlation coefficient). Overall, high consistency in the use of cues was observed, especially in Tempo ($\alpha$ = 0.950–0.957, calculated for the three subsets of the full design), Pitch ($\alpha$ = 0.928–0.936), and Mode ($\alpha$ = 0.894–0.940). The other cues also had high consistency, Articulation ($\alpha$ = 0.880–0.899), Brightness ($\alpha$ = 0.817–0.900), Dynamics ($\alpha$ = 0.799–0.849), with Instrumentation ($\alpha$ = 0.784–0.841) having the lowest consistency.

An initial exploration of the relationship between the cues and the factors Emotion, Piece, and the factors' interaction (Emotion x Piece) was carried out. First, a linear mixed model (LMM) was applied for each cue individually (using the *lmer* function from the *lme4* package in R), with Participant as the random factor (e.g., the base model for the tempo cue was as follows: Tempo ~ 1 + (1|Participant)). The factors (Emotion, Piece, and Emotion x Piece) were individually added to the linear mixed model (e.g., Tempo ~ 1 + Emotion + (1|Participant)). Generalised linear mixed models (GLMMs) with a binomial distribution were used instead of LMMs for the mode and articulation cues due to their binary nature. Likelihood ratio tests were then run to evaluate whether any of the factors added a statistically significant contribution to the initial model. Table 1 presents the results from the likelihood ratio tests between the initial models and with the added factors. The model which included Emotion as a factor differed significantly from the initial model for each of the separate cues. The addition of the Piece factor was of statistical significance for all cue models except the ones pertaining to the

brightness and pitch cues. The variance in statistical significance of Piece in relation to the different cues might suggest that the cues were used differently across musical pieces when attempting to convey the same emotion. This is presumably due to the variance in the musical structure of the pieces, which consequentially might affect how the cues are used to portray the same emotion across different pieces. The Piece and Emotion interaction did not have a significant contribution to the models.

Since the main aim of this paper is to better understand the relative contribution of the cues and their combinations to each of the seven emotions, the rest of the analysis will focus on how the cues were used together to communicate the different target emotions. To investigate this, separate LMMs were then calculated for each target emotion with respect to the different cues (Cue ~ Emotion + (1|Piece) + (1|Participant), e.g., Tempo ~ Sadness + (1|Piece) + (1|Participant)). Piece and Participant were inputted in the models as random factors. As these scores will be compared to Experiment 2 results in a later section, standardised beta scores (Z-scores) were utilised in the calculations rather than the raw scores for easier comparison of the results. Results of all LMM computations are shown in Table 2.

The first seven columns in Table 2 represent the seven emotions investigated in this experiment. The cues' LMM estimates for all emotions are shown in the rows in Table 2. The first four rows of Table 2 display the LMM estimates for the continuous cues, tempo, pitch, dynamics, and brightness. The sign (+ or -) of these four cues indicate whether the cue values (via beta coefficients) were positive (+) or negative (-). For example, a positive value for tempo suggests a fast tempo, and a negative value for pitch suggests a low pitch level. Rows 5 to 9 in Table 2 represent the estimates for the discrete cues, articulation, mode, and instrumentation. Due to the categorical nature of the instrumentation cue, each instrument option (brass, strings, and woodwinds) was regarded separately for the analysis and thus, split into three different rows in Table 2. As the LMM estimates of the categorical cues do not represent absolute values, the sign (+ or -) for each of these cues has different meanings. A negative value for the articulation indicates a smooth playing method (*legato*), whilst a positive value indicates a detached playing method (*staccato*). Minor mode is represented by a positive value, whilst a negative value represents major mode. A significant negative value for an instrument indicates that the instrument was specifically not chosen for the intended emotion. A significant positive value indicates that the instrument was explicitly chosen for the particular emotion. A non-significant value for any of the instruments suggests that the specific instrument did not play a role in the communication of the particular emotion.

**Table 1. The Chi-squared statistics ($\chi^2$) produced from likelihood ratio tests for separate G/LMM models of tempo, articulation, mode, pitch, dynamics, brightness, and instrumentation cues with and without the factors emotion, piece, and emotion x piece interaction.**

|  | Emotion | Piece | Emotion x Piece |
|---|---|---|---|
| **Tempo** | 677.04*** | 31.30*** | 48.45 |
| **Articulation** | 369.18*** | 23.46*** | 34.32 |
| **Pitch** | 223.68*** | 1.92 | 47.75 |
| **Dynamics** | 580.69*** | 27.81*** | 28.27 |
| **Brightness** | 341.14*** | 6.65 | 27.99 |
| **Mode** | 426.42*** | 17.19** | 41.21 |
| **Instrumentation** | 216.77*** | 23.57*** | 44.81 |

Notes.

* $p < .05$

** $p < .01$

*** $p < .001$, df = 6 for Emotion, df = 6 for Piece, df = 36 for Emotion x Piece Interaction for the likelihood ratio test.

**Table 2. Linear Mixed Model (LMM) results for seven emotions across all cues in Experiment 1.** The numbers are standardised betas (z-scores) with their 2.5% and 97.5% confidence intervals shown in brackets.

| | Sadness | Joy | Calmness | Anger | Fear | Power | Surprise | Mean Pseudo $R^2$ |
|---|---|---|---|---|---|---|---|---|
| **Tempo** | -1.44*** | 0.58*** | -1.25*** | 0.89*** | 0.36*** | 0.38*** | 0.47*** | 0.0305 |
| | (-1.60, -1.27) | (0.39, 0.76) | (-1.42, -1.08) | (0.71, 1.07) | (0.17, 0.54) | (0.19, 0.56) | (0.28, 0.65) | |
| **Pitch** | -0.75*** | 0.83*** | -0.28** | -0.19* | -0.41*** | 0.05 | 0.75*** | 0.0062 |
| | (-0.93, -0.57) | (0.65, 1.01) | (-0.46, -0.09) | (-0.38, 0.00) | (-0.59, -0.22) | (-0.14, 0.23) | (0.57, 0.93) | |
| **Dynamics** | -1.03*** | 0.25** | -1.22*** | 0.64*** | 0.21* | 0.80*** | 0.36*** | 0.0077 |
| | (-1.19, -0.87) | (0.07, 0.42) | (-1.38, -1.07) | (0.47, 0.81) | (0.04, 0.39) | (0.63, 0.97) | (0.19, 0.53) | |
| **Brightness** | -1.00*** | 0.72*** | -0.72*** | 0.17 | -0.34*** | 0.56*** | 0.62*** | 0.0057 |
| | (-1.17, -0.83) | (0.54, 0.90) | (-0.90, -0.55) | (-0.02, 0.35) | (-0.52, -0.16) | (0.38, 0.74) | (0.44, 0.80) | |
| **Articulation** | -3.58*** | 0.46* | -2.56*** | 1.00*** | 0.51* | 0.97*** | 2.04*** | 0.0103 |
| | (-4.54, -2.82) | (0.06, 0.87) | (-3.18, -2.01) | (0.58, 1.45) | (0.11, 0.93) | (0.55, 1.41) | (1.52, 2.63) | |
| **Mode** | 1.58*** | -3.10*** | -1.63*** | 2.32*** | 1.91*** | 0.28 | -1.71*** | 0.0397 |
| | (1.13, 2.07) | (-3.89, -2.43) | (-2.12, -1.18) | (1.78, 2.94) | (1.42, 2.45) | (-0.11, 0.68) | (-2.20, -1.24) | |
| **Brass Instrumentation** | 0.03 | 0.70*** | -2.16*** | 0.08 | -0.57* | 0.70** | 0.08 | 0.0089 |
| | (-0.43, 0.47) | (0.28, 1.11) | (-3.22, -1.34) | (-0.38, 0.52) | (-1.12, -0.07) | (0.28, 1.12) | (-0.38, 0.52) | |
| **Strings Instrumentation** | -0.89*** | -0.50* | -1.37*** | 1.26*** | 0.97*** | 0.58** | -0.07 | 0.0036 |
| | (-1.31, -0.48) | (-0.91, -0.11) | (-1.84, -0.94) | (0.83, 1.71) | (0.56, 1.40) | (0.18, 0.99) | (-0.46, 0.32) | |
| **Winds Instrumentation** | 0.94*** | -0.07 | 2.44*** | -2.87*** | -0.79** | -2.65*** | 0.02 | 0.0076 |
| | (0.54, 1.34) | (-0.52, 0.35) | (1.99, 2.91) | (-4.07, -1.97) | (-1.33, -0.30) | (-3.84, -1.76) | (-0.41, 0.44) | |
| **Pseudo $R^2$ (marginal)** | 0.33 | 0.19 | 0.36 | 0.20 | 0.12 | 0.11 | 0.15 | |

With regards to cue combinations for the individual emotions, the LMM estimates indicate that a slow tempo, low pitch level, soft dynamics, low brightness level (i.e., a dull sound), *legato* articulation, and minor mode were specifically used to communicate *sadness* [10, 34, 37, 53]. Furthermore, the woodwinds ensemble was explicitly used to convey sadness, whilst the strings ensemble was specifically not used [11, 32, 54]. The brass ensemble was not a contributing factor when conveying sadness. All cues except for brass instrumentation had significant roles in the communication of joy through the musical pieces. A fast tempo, high pitch level, loud dynamics, rich bright sound, *staccato* articulation, major mode, and brass instrumentation were specifically used to communicate *joy* [7, 11, 21, 32], whilst strings instrumentation was explicitly not used to convey joy. The woodwinds instrumentation did not have a significant effect on the portrayal of joy.

A slow tempo, low pitch level, soft dynamics, low brightness level (dark timbre), *legato* articulation, major mode, and woodwinds instrumentation significantly portrayed *calmness* [11, 32, 34, 43, 55]. Brass and strings instruments were specifically not chosen to convey calmness. All cues except for brightness and brass instrumentation had a significant impact in communicating anger. A fast tempo, low pitch level, loud dynamics, *staccato* articulation, minor mode, and strings instrumentation were specifically used to express *anger* in the musical pieces [21, 27, 29, 43]. The woodwinds ensemble was explicitly not chosen to portray anger.

Fear was communicated very similarly to anger; however, all cues significantly contributed to the intended emotion. A fast tempo, low pitch level, slightly loud dynamics, dark timbre, *staccato* articulation, minor mode, and strings instrumentation were used to convey *fear* [20, 21, 27, 29], whilst brass and woodwinds instrumentations were specifically not chosen when portraying fear. Tempo, dynamics, brightness, articulation, and instrumentation contributed to conveying power, whilst pitch and mode did not have a significant effect in conveying power through the music. A fast tempo, loud dynamics level, bright sound, *staccato* articulation, and brass and strings instrumentation were particularly chosen to help express *power* in

the musical pieces [27, 56]. The woodwinds ensemble was specifically not used to convey power. Lastly, the combination of fast tempo, high pitch level, loud dynamics, bright sound, *staccato* articulation, and major mode were all significant contributing factors to the communication of *surprise* [27, 34]. The instrumentation cue as a whole (i.e., all of brass, strings, and woodwinds ensembles) did not play a role in conveying surprise.

To further explore how effective the cue-emotion models (i.e., the cue combinations used per emotion) were to convey the intended emotions, the Pseudo $R^2$ value was calculated for each emotion, using the *r.squaredGLMM* function from the *MuMIn* package in R [57–59]. The Pseudo $R^2$ values present the proportion of variability of the emotion explained by the independent variables in the model (i.e., the cue combinations for each emotion), and these can be seen in the last row of Table 2. Calmness and sadness cue-emotion models had the highest Pseudo $R^2$ values of 0.36 and 0.33, respectively. Power (0.11) and fear (0.12) had the lowest Pseudo $R^2$ values, which suggests that the cue combinations used for these two emotions had the least stable emotion-cue models, compared to the others.

Finally, to investigate the impact of the individual cues on the portrayal of the different emotions, each cue's mean Pseudo $R^2$ value across all emotions was computed. These values are displayed in the last column of Table 2. It is to be noted that the three instrument ensembles (brass, strings, and woodwinds) together make up the instrumentation cue. Thus, to get an accurate mean Pseudo $R^2$ value, the individual mean Pseudo $R^2$ values are summed together. Mode (0.0397) and tempo (0.0305) provide the greatest contributions to communicate specific emotions, which is in line with previous findings [11, 29, 34]. The summation of the instrumentation cue (0.0201) is the third greatest contributor to emotional expression, followed by articulation. Brightness (0.0057) and pitch (0.0062) seem to have the least effect on how emotions are shaped in music.

In summary, the seven cues in question and their combinations mostly had a significant role in portraying all emotions targeted. Calmness and sadness had the highest overall prediction rates, whereas power and fear had the lowest associations between the cues and emotion via the models. As individual cues, mode and tempo had the heaviest contributory weight on how different emotions were shaped in the music, while brightness and pitch were, overall, the least contributing factors in shaping and communicating specific emotions through music.

## Experiment 2: Perception approach

A factorial design was used to systematically manipulate and render variations of the musical stimuli with different cue levels and combinations which were then evaluated in a perception study. The different cue levels were determined based on results from a previous study [34] which utilised the same musical stimuli and identified the relative optimal levels of tempo, pitch, dynamics, brightness, articulation, and mode for each of the musical stimuli to convey the seven different emotions (sadness, joy, calmness, anger, fear, power, and surprise). These optimal cue levels were utilised as the point of reference and mid-levels for the cue levels in this experiment. The rationale behind the other cue levels will be detailed in the Cue Details section (section 3.1.2).

### Method

A perception study was designed where the same seven musical cues studied in Experiment 1 were manipulated on two to three levels across seven different musical pieces.

**Stimulus manipulation.** Seven cues with 2–3 levels creates a large design matrix in terms of all possible cue and level combinations ($3 \times 2 \times 2 \times 3 \times 3 \times 3 \times 3 = 972$). Due to the size of the design, it would not be feasible for participants to respond to all possible trials. Therefore,

we calculated the number of trials needed to provide an optimal geometrical design where every main effect and first-order interactions would be balanced (e.g., design symmetry close to 1), without participants having to carry out all 972 trials. Using 36 trials out of the 972 for each musical piece gives a geometrical design symmetry of 0.956. In total, the fractional factorial design holds 252 different combinations of cue levels across the seven musical pieces (36 trials x 7 musical pieces = 252 combinations).

**Cue details.** *Tempo*. Three levels were computed for the tempo cue. Previous perception studies [11, 60, 61] either calculated different levels of notes per second (NPS) across musical pieces or used quantiles to determine different levels which cover a substantial range. In this experiment, the 0.35 and 0.65 quantiles were calculated with respect to the optimal cue level used as the mid-point (0.50 quantile).

*Brightness*. Similar to the tempo cue, three brightness levels were computed, utilising the quantiles of 0.35, 0.50, 0.65, relative to the optimal cue levels identified in a previous study [34].

*Dynamics*. Three levels of dynamics were computed for this study. Previous studies have utilised a dynamic range of 20dB [11, 32], representing the normal range of an acoustic instrument [11, 62]. Step-sizes were made in 5dB from -10dB to +10dB, and the dynamics controlled the sample synthesizer rather than the volume output. The stimuli to be utilised in this current study have specific dynamic values attributed to them, which already have been validated by two sets of participants [34]. Therefore, the pre-established dynamics values of the musical pieces were utilised as the calibrated mid-point and then two levels at a ±5dB difference were calculated.

*Pitch*. Three levels of pitch were calculated. Previous studies explored different pitch levels an octave (12 semitones) apart [32, 39], as well as seven semitones apart [11]. In this experiment, pitch levels at ±7 semitones from the point of reference were investigated. The register ranges of the instruments utilised in this experiment and the pitch range of the musical pieces to be used were taken into consideration when deciding on the different pitch levels, ensuring that all pitch levels chosen were in range. The only exception to this methodology was the musical piece intending to convey anger, since it originally had a low pitch range. As it would have been impossible for the instruments to play an octave lower than the original pitch, this was used as the lower pitch level, while the higher two levels were computed as +7 semitones and +14 semitones, respectively.

*Mode*. Two categorical levels of mode were chosen: major and harmonic minor. This cue was controlled through the 'Transposer' plug-in in Logic Pro X, the software that was used to render the stimuli for this experiment. Changes to the third and sixth scale degree from minor (m3/m6) to major (M3/M6) were made in the plug-in, to switch from minor and major mode, respectively.

*Articulation*. Two levels of articulation were investigated: *legato* and *staccato*. This cue was controlled through the virtual instrument plug-in (Vienna Symphonic Library) used to export the musical stimuli.

*Instrumentation*. The instrument groups used were the same as the ones available in *Emote-Control* V2.0 and used in Experiment 1, due to their expressive ranges and varied timbral sound. Furthermore, the use of the same instruments ensured consistency across the two experiments. The brass ensemble consisted of a piccolo trumpet, Vienna horn, euphonium, and trombone. The strings ensemble consisted of a violin, viola, cello, and double bass. The woodwinds ensemble was made up of a flute, clarinet, oboe, and bassoon.

**Creation of stimuli.** The seven musical pieces utilised in Experiment 1 served as the point of departure for the stimuli in this experiment. Thirty-six iterations for each original musical piece with different cue levels were exported. The Vienna Symphonic Library (VSL) was utilised as the virtual instrument in Logic Pro X to export all tracks.

**Procedure.**   Full ethical consent was approved by the Ethics Committee of Durham University (*MUS-2020-05-14T12:27:22-cfsg56*). The experiment was administered online using the survey tool Qualtrics. At the start of the study, participants were given a detailed description of the study and they provided written, informed consent. The 36 variations for each of the seven musical pieces were put in separate blocks in Qualtrics. To minimise fatigue, each participant was randomly presented two out of the seven blocks, which meant that each participant listened to 72 trials (all variations of two different pieces) out of the 252 total combinations. For each piece, participants rated how much of each of the seven emotions joy, sadness, calmness, power, anger, fear, and surprise they thought the music was conveying. Ratings were done on seven separate five-point Likert scales. A rating of 1 (none at all) indicated that the music did not convey any emotion. A rating of 5 (a lot) indicated that the music strongly conveyed the emotion. Participants carried out a practice trial to familiarise themselves with the music listening task and rating scales. The study took approximately 50 minutes to 1 hour to complete. Participation in this experiment was voluntary. Participants could opt-in a prize draw for two £10 Amazon vouchers at the end of the experiment.

**Participants.**   Participants were recruited via social media and university communications. 162 participants (51 men, 110 women, one individual preferred not to say) between 18 and 66 years (*M* = 34.22, *SD* = 13.05) took part in the study. A one-question version of the OMSI [48, 49] was utilised to distinguish between the participants' levels of musical expertise. Seventy-four of the participants were musicians, and 88 were non-musicians.

## Results

The consistency of the participants' ratings was calculated by examining the inter-rater agreement (using Cronbach's alpha) within each emotion scale across each participant and musical piece. High consistency was observed for all rating scales, especially in the sadness rating scale (α = 0.953), joy rating scale (α = 0.944), and calmness rating scale (α = 0.937). The other rating scales also had high consistency (fear α = 0.909, surprise α = 0.906, power α = 0.864), with the anger rating scale having the lowest consistency score α = 0.842.

Similar to Experiment 1, a linear mixed model (LMM) was applied for each target emotion with respect to the different cues to identify whether the cues and their combinations had a significant role in conveying the different emotions (Rating ~ Tempo + Pitch + Mode + Dynamics + Brightness + Articulation + Strings + Woodwinds + Brass + (1|Piece) + (1|Participant)). Participant and Piece were used as random factors in this analysis. Standardised scores (Z-scores) were utilised in the calculations, and the LMM estimates are shown in Table 3. The results should be interpreted in the same way as Table 2 (e.g., a negative value for articulation indicates *legato*). For a detailed explanation of how the LMM results in Table 3 should be interpreted, please see the explanation given in the Results Section of Experiment 1 for Table 2.

All cues except for the brass instrumentation had a significant effect in conveying sadness. A slow tempo, low pitch level, soft dynamics, dull sound (low brightness level), *legato* articulation, minor mode, and strings instrumentation were specifically used to convey *sadness* [7, 10, 14, 20, 24, 27, 29, 34, 63, 64]. The woodwinds ensemble was explicitly not used to convey sadness, whilst the brass ensemble was not a statistically significant contributing factor to the communication of sadness. A fast tempo, high pitch level, bright sound, *staccato* articulation, major mode and woodwinds instrument were used to portray *joy* [7, 27, 32, 54, 65]. The strings instrumentation was specifically not chosen to portray joy through music, while the dynamics cue and brass instrumentation did not contribute to the intended emotion. A slow tempo, low pitch level, low brightness level, *legato* articulation, major mode, and brass instrumentation were specifically chosen to communicate *calmness* [32, 35, 43]. The woodwinds

ensemble was explicitly not chosen, while dynamics and strings instrumentation were not contributing factors towards calmness. *Anger* was portrayed by a fast tempo, low pitch level, *staccato* articulation, minor mode, and strings instrumentation [6, 21, 27, 35]. The woodwinds ensemble was specifically not chosen to convey anger, whilst dynamics, brightness, and brass instrumentation did not help to express anger through the musical pieces.

A fast tempo, low pitch level, minor mode, and strings instrumentation were strategically utilised to convey the emotion *fear* [11, 20, 21, 27, 32], whilst brass and woodwinds instrumentations were specifically not chosen. The dynamics, brightness, and articulation cues did not have a significant role in conveying fear through the music. Tempo, pitch, articulation, mode, and instrumentation contributed to conveying *power*, whilst dynamics and brightness did not have a significant effect on communicating power. A fast tempo, low pitch level, *staccato* articulation, major mode, and strings instrumentation were specifically chosen when power was being conveyed [27]. The brass and woodwinds ensembles were explicitly not used, whilst dynamics and brightness did not significantly affect the portrayal of the power. All cues except for dynamics and mode had a significant effect on the communication of the surprise emotion. A fast tempo, high pitch level, bright sound, *staccato* articulation [27, 34], and woodwinds instrumentation contributed to the conveying of *surprise* in the musical pieces, whilst brass and strings instrumentations were explicitly not used.

The Pseudo $R^2$ marginal values were also computed to investigate how well the cue combinations used could accurately predict the intended emotions. The cue-emotion models for sadness (0.21) and joy (0.20) had the highest Pseudo $R^2$ values, whilst the cue-emotion models for power (0.04) and anger (0.06) had the lowest scores.

Lastly, to determine the effect size of each cue on the shaping of the emotions, the mean Pseudo $R^2$ value for each cue across all emotions was computed. These values are displayed in the last column of Table 3. Mode (0.0561) and articulation (0.0373) had the biggest effect on the shaping of the different emotions, followed by tempo (0.0136) and the instrumentation cue as a whole (0.0104). Dynamics (0.0002) and brightness (0.0007) were the least effective cues on the emotion expressed in the music. This can also be seen from the LMM estimates of dynamics and brightness in Table 3, where dynamics had a significant effect only for sadness, and brightness significantly affected only four (sadness, joy, calmness, and surprise) of the seven expressed emotions.

In summary, most of the seven cues investigated in this perception experiment had a significant role in portraying sadness, joy, calmness, anger, fear, power, and surprise. The dynamics cue only had a significant effect on the conveying of sadness from the seven target emotions. As individual cues, mode and articulation had the biggest effect on modelling the desired emotion, while dynamics and brightness were the least contributing factors in shaping specific emotional expressions through music.

## Comparison of Experiments 1 and 2

Table 4 presents a high-level overview of cue levels used to portray the different emotions in Experiment 1 (the production study) and Experiment 2 (the perception study). Cue levels in a bold font indicate results which adhere to findings from previous literature. Overall, the cues operate similarly in the majority (32/49) of cue-emotion combinations across the two experiments. Only five cue-emotion discrepancies are related differences where both cues are statistically significant and different (as in tempo and anger, +/~). Additionally, none of the continuous cues demonstrate opposite cue values as all discrepancies are a matter of nuance (as in tempo and anger, where high tempo levels were associated with anger in Experiment 1 but medium levels in Experiment 2). This is not entirely surprising as the two experiments are

**Table 3. Linear Mixed Model (LMM) results for seven emotions across all cues in the perception experiment.** The numbers are standardised betas, and their 2.5% and 97.5% confidence intervals are denoted in brackets.

| | Sadness | Joy | Calmness | Anger | Fear | Power | Surprise | Mean Pseudo $R^2$ |
|---|---|---|---|---|---|---|---|---|
| **Tempo** | -0.21*** | 0.16*** | -0.22*** | 0.03** | 0.05*** | 0.11*** | 0.08*** | 0.0136 |
| | (-0.23, -0.18) | (0.13, 0.18) | (-0.25, -0.20) | (0.01, 0.05) | (0.03, 0.08) | (0.08, 0.13) | (0.06, 0.11) | |
| **Pitch** | -0.14*** | 0.17*** | 0.04*** | -0.10*** | -0.06*** | -0.09*** | 0.07*** | 0.0074 |
| | (-0.16, -0.11) | (0.14, 0.19) | (0.02, 0.06) | (-0.12, -0.08) | (-0.08, -0.03) | (-0.12, -0.07) | (0.05, 0.10) | |
| **Dynamics** | -0.03** | -0.02 | -0.02 | 0.00 | 0.02 | 0.02 | -0.01 | 0.0002 |
| | (-0.05, 0.01) | (-0.04, 0.01) | (-0.04, 0.00) | (-0.02, 0.02) | (-0.01, 0.04) | (-0.01, 0.04) | (-0.03, 0.02) | |
| **Brightness** | -0.02* | 0.05*** | -0.05*** | 0.01 | -0.02 | -0.02 | 0.04*** | 0.0007 |
| | (-0.05, 0.00) | (0.03, 0.07) | (-0.07, -0.02) | (-0.01, 0.03) | (-0.04, 0.01) | (-0.04, 0.01) | (0.02, 0.07) | |
| **Articulation** | -0.44*** | 0.14*** | -0.27*** | 0.06*** | 0.01 | 0.10*** | 0.30*** | 0.0373 |
| | (-0.46, -0.41) | (0.11, 0.16) | (-0.29, -0.24) | (0.04, 0.08) | (-0.01, 0.04) | (0.08, 0.13) | (0.28, 0.33) | |
| **Mode** | 0.27*** | -0.50*** | -0.24*** | 0.20*** | 0.42*** | 0.04*** | -0.01 | 0.0561 |
| | (0.24, 0.29) | (-0.52, -0.47) | (-0.26, -0.22) | (0.18, 0.22) | (0.39, 0.44) | (0.02, 0.07) | (-0.03, 0.01) | |
| **Brass Instrumentation** | 0.01 | 0.06 | 0.08** | -0.04 | -0.12*** | -0.12*** | -0.09*** | 0.0026 |
| | (-0.05, 0.06) | (0.00, 0.12) | (0.03, 0.13) | (-0.08, 0.01) | (-0.18, -0.07) | (-0.17, -0.06) | (-0.14, -0.04) | |
| **Strings Instrumentation** | 0.10*** | -0.12*** | -0.03 | 0.11*** | 0.21*** | 0.33*** | -0.07* | 0.0038 |
| | (0.04, 0.15) | (-0.18, -0.06) | (-0.08, 0.02) | (0.06, 0.15) | (0.15, 0.26) | (0.28, 0.39) | (-0.12, -0.01) | |
| **Woodwinds Instrumentation** | -0.10*** | 0.06* | -0.05* | -0.07** | -0.08** | -0.22*** | 0.16*** | 0.0040 |
| | (-0.16, -0.05) | (0.00, 0.12) | (-0.10, 0.00) | (-0.11, -0.03) | (-0.14, -0.03) | (-0.27, -0.17) | (0.11, 0.21) | |
| **Pseudo $R^2$ (marginal)** | 0.21 | 0.20 | 0.15 | 0.06 | 0.12 | 0.04 | 0.08 | |

based on the same underlying musical pieces and the manipulation of the same cues, although the actual cue levels are not directly comparable. This internal consistency is nevertheless reassuring and may be interpreted as an internal validation of the approaches used.

When we compare the cue levels for the different emotions explored in this study to findings of past studies, this mostly supports the previous literature [1, 13, 66, 67]; A slow *tempo* is associated with sadness and calmness, a moderate or fast tempo is linked to anger and fear, and a fast tempo is associated with joy, power, and surprise [6, 7, 24, 29, 32, 35, 37, 43, 56, 63]. Furthermore, tempo had a significant effect in shaping all the different emotions in the music.

**Table 4. A summary of cue levels used to shape the different emotions in Experiment 1 and 2.**

| | Emotion | Sadness | Joy | Calmness | Anger | Fear | Power | Surprise |
|---|---|---|---|---|---|---|---|---|
| | Source | E1/E2 | E1/E2 | E1/E2 | E1/E2 | E1/E2 | E1/E2 | E1/E2 |
| **Cues** | Tempo | - / - | + / + | - / - | + / ~ | + / ~ | + / + | + / + |
| | Dynamics | - / ~ | + / [~] | - / [~] | + / [~] | + / [~] | + / [~] | + / [~] |
| | Pitch | - / - | + / + | - / ~ | - / - | - / - | [~] / - | + / + |
| | Brightness | - / - | + / + | - / - | [+] / [~] | - / [–] | + / [–] | + / + |
| | Articulation | **L / L** | **S / S** | **L / L** | **S / S** | **S** / [S] | **S / S** | **S / S** |
| | Mode | - / - | + / + | + / + | - / - | - / - | [+] / - | + / [+] |
| | Instrumentation | **W / S** | **B** / W | **W** / B | **S / S** | **S / S** | B, **S / S** | [] / W |

Notes. The cue levels of Experiment 1 are denoted first in each cell, followed by Experiment 2 values. Source of data is denoted by the source row, where E1 refers to Experiment 1 and E2 refers to Experiment 2. Results from the two experiments are separated by /, and symbols denote cue levels (+ high level, ~ mid-level,—low level). Cue levels all had a significant effect, except for ones in square brackets []. Articulation cue: L = *legato*, S = *staccato*. Mode cue:— = minor, + = major. Instrumentation cue: B = brass, S = strings, W = woodwinds, [] = none of the instrument ensembles were significant. Levels in a bold font indicate results that adhere to findings from previous literature.

The *dynamics* cue presented rather diverging results between the two experiments in this study. In Experiment 1, the dynamics cue was a significant contributing factor to all emotions, varying in importance. A low dynamics level was used for sadness and calmness, and a loud dynamics level was used for joy, anger, fear, power, and surprise, which overall complements the existing literature [7, 29, 32, 35, 55]. Dynamics had the least impact, albeit significant, in the communication of fear. Interestingly, in previous studies, it has been suggested that it is possible to convey fear with both a low dynamics level [54, 65] and a high dynamics level [11, 32].

On the other hand, in Experiment 2, the dynamics cue had the least effect on creating different emotion profiles in music. A low dynamics level was purposely used to portray sadness in Experiment 2, which complements previous research [11, 35, 39, 56]. However, dynamics did not have a significant effect on portraying any of the other emotions. The discrepancy between the non-significant effect of the dynamics cue in Experiment 2, the findings of Experiment 1, and previous research is rather notable. Existing literature provides evidence that different dynamics levels have an impact on the emotion being expressed by the music [13, 29, 30, 32, 35], where high activity emotions are usually associated with high dynamics levels, and low activity emotions with low dynamics levels [68]. The fact that the three different levels of the dynamics cue used in Experiment 2 were based on quantiles varying in increments of 0.15 means that differences between the levels would be rather subtle, as Experiment 2 focussed on fine-tuning cues, rather than using drastically different upper and lower limits. It is possible that participants did not distinguish between the minor changes between the dynamics cue levels [69], which might explain why overall, the cue did not register as having a significant impact on the emotional expression. Furthermore, Experiment 2 was online based, which is a less controlled experiment environment. Although the instructions informed participants to use headphones and set their volume to a comfortable level before the experiment, it is entirely possible that participants used less than adequate headphones, no headphones at all, or changed the volume of their device while carrying out the study. Any of these factors might have had an influence on the dynamics.

For the most part, the *pitch* cue was consistently used across both experiments and mostly had a significant effect on the conveyed emotion. A low pitch level was used in both experiments for sadness, anger, and fear [7, 11, 24, 32, 39, 55, 56]. A low pitch level was also purposely used for calmness in Experiment 1, while a moderate pitch level was preferred for calmness in Experiment 2. A high pitch level was utilised to convey joy and surprise in both experiments [11, 24, 27, 70, 71]. A low pitch level was explicitly used for power in Experiment 2, while the pitch level did not significantly affect power in Experiment 1. As with the dynamics cue, conflicting data on which pitch level conveys different emotions exists in current literature. For example, both high and low pitch levels have been used to communicate power, fear, calmness, and anger [6, 11, 27, 32, 34, 54].

In both Experiment 1 and 2, a low *brightness* level, i.e., one with few upper harmonics, which creates a dull sound, was used for sadness and calmness [50, 54, 65]. In Experiment 1, a low brightness level was also used to convey fear; however, in Experiment 2, brightness had a non-significant effect on the portrayal of fear. A high brightness level, i.e., one with multiple harmonics and a bright sound, was specifically used for joy and surprise in both experiments [7, 27, 54]. A high brightness level was also used in the portrayal of power in Experiment 1 [27], while Experiment 2 produced a non-significant brightness value. In both experiments, the brightness cue did not play a significant role in shaping anger. It has been noted that high brightness sounds are more likely associated with high activity emotions, while low activity emotions are more likely represented by dull and dark sounds [32, 72].

The *articulation* cue produced consistent results for all emotions across both experiments. A *legato* articulation was specifically chosen to portray sadness and calmness, while a *staccato* articulation was used to convey joy, anger, fear, power, and surprise [29, 32, 35, 53, 65], consistent with past findings. Furthermore, the articulation cue had a significant effect in conveying all emotions, except for fear in Experiment 2. *Mode* was also used similarly across both experiments and the current literature. A major mode was utilised to convey the positive emotions joy, calmness, and surprise, while a minor mode was chosen for sadness, anger, fear, and power emotions [10, 27, 34, 63, 73]. However, the mode did not significantly affect power in Experiment 1 and surprise in Experiment 2.

Finally, the *instrumentation* cue had quite contrasting results between the two experiments. The instrumentation cue was used similarly only for anger and fear, where a strings instrumentation was specifically chosen to portray the aforementioned emotions [20, 21, 43] in both experiments. Additionally, the woodwinds instrumentation was specifically not chosen for anger and fear, and brass was also specifically not used to portray fear. In Experiment 1, a woodwinds ensemble was specifically used to convey sadness [11, 32, 54], while a strings ensemble was distinctively not chosen. The opposite findings can be seen in Experiment 2, where a strings instrumentation was used for sadness [20, 51], and woodwinds instrumentation was specifically not used. Brass instrumentation was chosen for joy in Experiment 1 [21, 32], while a woodwinds ensemble was preferred in Experiment 2. Strings instrumentation was specifically not used to convey joy in both experiments. A woodwinds ensemble was utilised to convey calmness [11, 32, 43] in Experiment 1, with brass and strings instrumentations specifically not chosen to portray the aforementioned emotion.

On the other hand, a brass ensemble was utilised in Experiment 2, and a woodwinds ensemble was specifically not chosen to portray calmness. A strings instrumentation was preferred for power in both experiments. Additionally, a brass ensemble was also chosen by participants to represent power in Experiment 1, while brass was specifically not chosen in Experiment 2. A woodwinds instrumentation was specifically not used to portray power in both experiments. A woodwinds instrumentation was specifically utilised for surprise in Experiment 2, while brass and strings ensembles were purposely not chosen to portray surprise.

The instrument and emotion association results produced in this paper partially adhere to a handful of studies that looked at individual instruments rather than instrument ensembles [11, 32, 51, 54]. However, there are also conflicting results both between the findings of the two current experiments and other previous findings. For example, it has also been reported that sadness is best represented with a trumpet [21] or French horn [32], and that voice (not investigated here) and strings instruments, such as violin and cello, might have a bigger sadness capacity than other instruments [20, 51]. Balkwill and Thompson [43] reported that the instrument timbre did not have a significant effect on the portrayal of sadness and joy in music. Other studies have reported that fear is best conveyed with a brass instrument, such as the French horn [32], or potentially a trumpet [11]. The fact that we used instrument ensembles rather than individual instruments should also be taken into consideration. Although the instrumentation results have been compared to previous findings involving individual instruments from the same instrument families (e.g., a trumpet compared to the brass ensemble), our instrumentation cue consists of instrument combinations rather than individual instruments, which might also affect the perceived emotion. Additionally, most previous studies have investigated the effect of instrument timbre and other cues on monophonic melodies, while this current study explores the effect of instrument ensembles and the other cues on the overall structure of polyphonic music. It has been suggested that music with multiple musical parts likely provides more information than monophonic melodies [21, 74].

It is also worth noting that brightness is one of the major perceptual dimensions of timbre [52, 75]. Saitis and Siedenburg [52] found that brightness perception dissimilarity is not distinguished by source-cause categories, i.e., different instrument families. Thus, altering the brightness component together with the instrumentation cue in this work might have affected how participants used these two cues to portray the different emotions. Due to the intrinsic relationship between brightness and timbre, future studies should expand research on how the potential connection between instrument combinations and brightness impacts the perceived emotional expression in music.

The impact of the individual cues across emotions was ranked similarly in Experiments 1 and 2. Pseudo $R^2$ values show that overall, mode had the highest impact on the different emotion profiles in both experiments (Experiment 1 Pseudo $R^2$ = 0.0397, Experiment 2 Pseudo $R^2$ = 0.0561). In Experiment 1, mode was followed by tempo as the second most influential cue (Pseudo $R^2$ = 0.0305), which is consistent with the results from two previous studies [11, 34]. In the existing literature, it has also been suggested that tempo has the greatest impact on emotion shaping [27, 29]. The other cues in Experiment 1 were ranked as follows: instrumentation (Pseudo $R^2$ = 0.0201), articulation (Pseudo $R^2$ = 0.0103), dynamics (Pseudo $R^2$ = 0.0077), pitch (Pseudo $R^2$ = 0.0062), and brightness (Pseudo $R^2$ = 0.0057). In Experiment 2, articulation (Pseudo $R^2$ = 0.0373) scored as the second most impactful cue on the expressed emotion, followed by tempo (Pseudo $R^2$ = 0.0136), instrumentation (Pseudo $R^2$ = 0.0104), pitch (Pseudo $R^2$ = 0.0074), brightness (Pseudo $R^2$ = 0.0007), and dynamics (Pseudo $R^2$ = 0.0002). Although the cue impact ranking varies slightly between the two experiments, it is overall quite similar. In both experiments, mode had the most effect on shaping different emotions, followed by tempo, instrumentation, and articulation. In both experiments, pitch, dynamics, and brightness had the lowest scores, with a distinct difference between them and the first four cues. Differences in dynamics and brightness might be the least perceptive in an online music listening environment, which might explain the low effect scores of dynamics and brightness in Experiment 2. It is interesting to note that mode, tempo, and articulation, i.e., the cues flagged as the ones mostly contributing to the communication of the different emotions [11], were the cues used most consistently in both experiments.

When we compare the Pseudo $R^2$ marginal values of the cue-emotion models, we find that overall, the cue-emotion profiles in Experiment 1 scored noticeably higher than the ones used in Experiment 2. This indicates that the cue-emotion models used by participants in the production study are able to capture the variation more fully than the cue-emotion models produced in the perception study. The fact that in Experiment 1, participants could explore a wider range of the continuous cues (tempo, pitch, dynamics, and brightness) than in Experiment 2 suggest that the methodological limitations inherent in the perception approach where cue combinations were systematically manipulated led to situations where the optimal cue-emotion patterns were probably not always within the pre-defined cue values used in Experiment 2. In Experiment 1, Pseudo $R^2$ values suggest that the cue patterns used for calmness (Pseudo $R^2$ = 0.36) and sadness (Pseudo $R^2$ = 0.33) were the ones most reliable in conveying the intended emotion, compared to the other investigated emotions; anger (Pseudo $R^2$ = 0.20), joy (Pseudo $R^2$ = 0.19), surprise (Pseudo $R^2$ = 0.15), fear (Pseudo $R^2$ = 0.12), and power (Pseudo $R^2$ = 0.11). These findings support a recent production study that identified calmness and sadness as the two emotions best predicted by a combination of tempo, pitch, dynamics, brightness, articulation, and mode [34], and fear and surprise having the least reliable cue-emotion patterns. In Experiment 2, sadness (Pseudo $R^2$ = 0.21) and joy (Pseudo $R^2$ = 0.20) had the highest Pseudo $R^2$ scores for cue-emotion model reliability, followed by: calmness (Pseudo $R^2$ = 0.15), fear (Pseudo $R^2$ = 0.12), surprise (Pseudo $R^2$ = 0.08), anger (Pseudo $R^2$ = 0.06), and power (Pseudo $R^2$ = 0.04).

Across the two experiments, sadness, calmness, and joy emotions seem to have the most reliable cue combinations in shaping the intended emotion. Previous research has proposed that basic emotions, i.e., sadness, happiness, anger, fear, surprise, and disgust [76], are easier to communicate in music than other emotions [44, 60, 65, 77], potentially due to similarities in how basic emotions are expressed in vocal expression and music performance [5]. Although sadness and joy (basic emotions) had two of the highest predictive model accuracy scores [35] in this paper, this theory does not explain the high ranking of the cue-emotion model for calmness, which is not considered a basic emotion, and how it surpassed other basic emotions, such as anger and fear, which had a high identification rate in previous studies [26, 43, 78]. Interestingly, sadness, joy, and calmness have been reported as being three of the emotions most often attributed to music [1, 2, 61, 79, 80]. An alternative theory to the current findings is that music better expresses emotions, or rather, affective states, that can be explained without having a particular intent, unlike other emotions such as disgust, that are experienced in a specific, intentional situation context [67].

The comparison between Experiments 1 and 2 and the existing literature shows how overall, similar cue patterns were used to express specific emotions in music. The cues registered as the ones contributing most to the different emotion profiles (mode, tempo, and articulation) are used consistently to alter the emotional expression in music, while others sometimes varied between studies, such as the dynamics cue having a significant effect on all emotions in Experiment 1 but only having a significant effect on the portrayal of sadness in Experiment 2. In addition, the fact that different cue values have been reported as significantly affecting the emotional expression suggests that the cue values used are relative to the other cues being used to convey said emotion. Thus, this study provides evidence that supports the notion that cues work together to communicate the different emotions in their relative context [2, 10–12]. For example, overall, mode had the highest impact on the emotion profiles; however, it did not significantly affect the portrayal of power in Experiment 1. Instead, the power emotion profile was holistically built using the combination of cues.

Most importantly, unlike previous studies exploring a single cue [14, 15, 21, 23, 25], or a restricted amount of cues and levels, this study extensively explores the effect of a combination of seven musical cues and their multiple cue levels on the emotion expressed in the music, providing insights into the merits of these two methodological approaches.

## Discussion

This paper has provided two sets of data investigating the same seven musical cues in relation to seven emotional expressions, using two different approaches: production and perception. The findings support the notion that musical cues and their additivity communicate distinct emotions in music [3, 5, 81]. Overall, five out of the seven cues (tempo, pitch, brightness, articulation, and mode) were utilised in the same manner across the two experiments. The dynamics cue varied between the two experiments, as it significantly contributed to all emotions in Experiment 1 but was not significant for all but one emotion (sadness) in Experiment 2. The instrumentation cue produced the most variance between the two experiments, where the instrument of choice was similar for three (anger, fear, and power) out of the seven emotions.

Although the two approaches mostly produced similar findings, the differences between the obtained results raise the question of which approach could be deemed more useful. For example, the dynamics cue was the weakest contributing factor (Pseudo $R^2$ = 0.0002) in the perception experiment (Experiment 2). This might be because the differences between the dynamics levels were subtle and perhaps could not be perceived by the participants [69], especially in an online study, where the experiment environment cannot be fully controlled [82]. Therefore,

perhaps a different methodological approach that allows more control over the dynamics cue and research environment might be better suited when investigating dynamics.

Both approaches have advantages, as well as limitations. The production approach allows for participant engagement and direct user experience, where participants have the opportunity to *show* us how they would change the music to express the intended emotions. Furthermore, the production approach allows for a substantially large cue space to be explored in a relatively short time, which would not be possible with a perception study utilising a systematic manipulation design, where all cue combinations would have to be pre-defined, rendered, and listened to by participants [47]. One downside to giving the participants free rein of the cue space is the possibility that certain cue levels and combinations might not be explored. Another limitation of the production approach is that it was designed as a lab experiment, where the researcher could have full control of the research environment. This is a limitation that might not necessarily be that restrictive usually. However, due to the COVID-19 pandemic, face-to-face lab experiments were either not possible or limited, and thus, online methodologies that do not require physical contact would be ideal. A solution for this would be adapting the computer interface to an online setting.

One of the biggest advantages of a perception approach is that the researcher has total experimental control on the cue combinations explored. This allows studying small differences between cue levels, which might not be explored by participants using a production task. Perception experiments can easily be administered online, making them readily accessible and available to a larger population. Additionally, using online crowdsourcing platforms might make recruiting more diverse samples easier [83]. The downside of a perception approach is that it is unfeasible to investigate many cue combinations and levels simultaneously, as it easily leads to participant fatigue and lack of engagement [84]. There is a possibility of optimising the comparison through fractional factorial designs and dividing subsets of the stimuli across the participants in an optimal fashion, but even these techniques will not remove the combinatorial problems inherent in this approach.

In summary, this paper showed how combinations of seven musical cues shaped seven different emotion profiles in music, across two different methodological approaches used in cue-emotion research. Furthermore, utilising both production and perception approaches to investigate the same cues and emotions across studies allowed for a critical evaluation of methodological approaches used for this purpose and confirmed that similar cue-emotion patterns were discovered overall. This is particularly meaningful as it suggests that using *EmoteControl* as part of a production approach to musical cues and emotion research may be suitable, since similar results to the perception study were obtained. Moreover, the production approach created cue-emotion models that had a higher predictive model accuracy score than the ones produced in the perception study. The production task was also quicker to administer.

Most importantly, this work explored cue-emotion mapping utilising an ambitious number of cues simultaneously and exceeded past research on the cue-emotion space by using continuous cues with wide ranges and categorical cues with multiple levels. This gives us a glimpse of how real-time interactive approaches [6, 7, 32–35] may be used to explore numerous, complex cue-emotion mappings that exist in real music. Although the findings from the two experiments in this paper have shown us that overall, similar results in cue-emotion research are achieved with the two different approaches, only a sliver of the cue-emotion space can be realistically explored simultaneously using a traditional, perception approach, like the one used in Experiment 2. In the production study detailed in this paper, participants navigated through a possible two billion distinct points in the cue-emotion space in real-time (the number of iterations was calculated by multiplying all possible levels of tempo, mode, articulation, pitch, brightness, dynamics, and instrumentation cues respectively: 171 x 2 x 2 x 128 x 80 x 100 x

3 = 2,101,248,000), whilst 252 cue combinations were explored in the perception study. Furthermore, real music consists of a considerably larger number of cues and their combinations than explored here and is more complex than the reduced music samples used in perception studies [4]. Thus, the findings of this paper suggest that production studies (analysis-by-synthesis) offer a promising way forward in uncovering how the cue-emotion space operates in real music.

Moreover, the use of interactive paradigms would allow to investigate cue usage in relation to emotional expressions across different population samples, if the paradigm is easy to use and does not require any particular expertise. For example, Saarikallio et al. [6] and Kragness et al. [85] have already successfully utilised interactive paradigms to investigate how children use three musical cues to communicate a small selection of emotions in music. Another potential avenue would be to explore whether cue usage in portraying different emotions varies depending on musical expertise. Some literature has reported that musical expertise may affect a listener's decoding accuracy of perceived emotions in music [54, 86, 87]. However, Kragness and Trainor [35] reported that musical expertise had no distinct significant effect on how participants used a selection of cues in an interactive paradigm to 'perform' different emotions in chords from Bach chorales. Thus, probing this line of enquiry from a production paradigm's perspective would provide more data on whether musical expertise plays a role in how emotions are perceived in music. Additionally, employing an interactive production approach may also be beneficial in other research areas, such as ethnomusicology. For example, Arom, Léothaud, and Voisin [88] employed an interactive experimental procedure to investigate the musical scales and pitches used in Central Africa and Java. They created a device linked to a digital synthesiser that could simulate different traditional instruments. The researchers then asked native musicians, instrument makers, and tuners to retune the synthesised simulations of their traditional instruments by altering pitches on the device, to determine the scales and intervals used in traditional music of Central Africa and Java.

In conclusion, future studies investigating musical cues and emotion should move away from perception approaches which restrict them to a finite number of cue combinations. Instead, they should focus on alternative ways that tackle the large cue space more efficiently and further expand the cue-emotion space investigated to include other features that contribute to the emotional expression in music, such as harmony, sound space, articulation, other musical structural organisation principles than the current programming of the mode cue (e.g., adding other types of modes, tunings, etc.), panning of sound, and other aspects of timbre, such as spectral shape. It would be worthwhile to explore the variation in the cues across musical genres and traditions. Apart from production approaches that use computer interfaces or slider apparatus [6, 7, 32, 33, 47], other techniques include the self-pacing method, which allows participants to express emotions in music using a number of expressive cues [35, 36], the Markov Chain Monte Carlo with People (MCMCP) randomised algorithm which is used to understand participants' representations of perceptual objects and predict their behaviour [89] or Gibbs Sampling with People which uses a continuous-sampling paradigm of MCMCP and investigates how participants optimise cues for different emotional expressions [90].

## Supporting information

**S1 File. Pilot experiment.**
(DOCX)

**S2 File. Experiment 1 safety measures.**
(DOCX)

## Acknowledgments

The authors wish to thank all the participants who took part in the experiments, and the reviewers for their invaluable feedback and suggestions.

## Author Contributions

**Conceptualization:** Annaliese Micallef Grimaud, Tuomas Eerola.

**Data curation:** Annaliese Micallef Grimaud, Tuomas Eerola.

**Formal analysis:** Tuomas Eerola.

**Investigation:** Annaliese Micallef Grimaud.

**Methodology:** Annaliese Micallef Grimaud, Tuomas Eerola.

**Project administration:** Annaliese Micallef Grimaud.

**Software:** Annaliese Micallef Grimaud.

**Supervision:** Tuomas Eerola.

**Writing – original draft:** Annaliese Micallef Grimaud, Tuomas Eerola.

**Writing – review & editing:** Annaliese Micallef Grimaud, Tuomas Eerola.

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
