## [Decision Letter · Decision Letter 0]

9 Nov 2022

PONE-D-22-22560Emotional expression through musical cues: A comparison of production and evaluation approachesPLOS ONE

Dear Annaliese Micallef Grimaud, PhD Candidate

Thank you for submitting your manuscript to PLOS ONE. After careful consideration, we feel that it has merit but does not fully meet PLOS ONE’s publication criteria as it currently stands. Therefore, we invite you to submit a revised version of the manuscript that addresses the points raised during the review process.

ACADEMIC EDITOR: Thank you for submitting this well-designed study that reinforces existing knowledge about music and emotion with some details. I fully agree with the constructive criticism and guiding comments made by the reviewers by carefully examining your article and would like to point out that it requires some minor revisions in this context.

I look forward to receiving your revised article, which you will prepare in line with the recommendations of the reviewers.

We look forward to receiving your revised manuscript.

Kind regards,

Sukru Torun

Academic Editor

PLOS ONE

Journal Requirements:

2. lease change "female” or "male" to "woman” or "man" as appropriate, when used as a noun (see for instance https://apastyle.apa.org/style-grammar-guidelines/bias-free-language/gender)."

3. Please ensure that you have specified (1) whether consent was informed and (2) what type you obtained (for instance, written or verbal, and if verbal, how it was documented and witnessed). If your study included minors, state whether you obtained consent from parents or guardians. If the need for consent was waived by the ethics committee, please include this information.

Reviewers' comments:

Reviewer's Responses to Questions

**Comments to the Author**

1. Is the manuscript technically sound, and do the data support the conclusions?

Reviewer #1: Yes

Reviewer #2: Yes

2. Has the statistical analysis been performed appropriately and rigorously? 

Reviewer #1: Yes

Reviewer #2: Yes

3. Have the authors made all data underlying the findings in their manuscript fully available?

Reviewer #1: No

Reviewer #2: Yes

4. Is the manuscript presented in an intelligible fashion and written in standard English?

Reviewer #1: Yes

Reviewer #2: Yes

5. Review Comments to the Author

Reviewer #1: I appreciated the opportunity to review “Emotional expression through musical cues: A comparison of production and evaluation approaches”. In this work, the authors directly compared the results of a perception vs. production approach to the study of cues underlying musical emotions. Overall, they found that generally the same cues were used in both perception and production of emotions. Based on this, they suggest that production may be a more efficient approach to the study of musical emotions, since multiple cues can be simultaneously incorporated.

In general, my impression of the work is positive. I especially enjoyed the discussion about how the production approach could be used to test different populations. It’s also great to see that the authors were able to implement the EmoteControl online, which opens the door to many exciting future studies.

There are several points that I would like to see addressed before I can confidently recommend the work for publication. I have some minor comments about how their findings are interpreted (see below), which I’m confident the authors will be able to address. My more major comment about the work is that it’s not clear to me how it offers a novel contribution to the overall discussion on this topic. At the same time, my position is that novelty is overrated in the publication process, and my preference as a scholar is to have as many well-conducted studies in the literature as possible, regardless of novelty. So I would like to hear more from the authors about the specific significance of this work, but I do not think that a high degree of novelty is necessary for publication of the work. Therefore, my recommendation is for the authors to undertake minor revisions.

- The main goal of the work seems to be to directly compare a perception approach vs. a production approach. As the authors acknowledge, there are still quite substantial (though necessary) differences between the two paradigms they use – different groups of participants, the number of levels available to the participants, etc. In this sense, I’m not totally sure how this approach offers anything unique above and beyond their review of the literature comparing perception and production studies that have already been conducted. Perhaps there is additional usefulness in using the same cues and the same excerpts? A clearer explanation of the benefits of this specific approach would be useful.

- I find the analyses somewhat confusing. My first confusion is that I’m not sure how to interpret the effects of Piece reported in Table 1 (or even why it’s included as a predictor in the first place). Does it mean, for example, that some pieces were just generally chosen to be faster than other pieces (presumably because compositional cues suggested different optimal tempos)? Would it make more sense to treat Piece as a random factor? I’m not suggesting it is necessary for the authors to do this; I’m simply trying to better understand how I should interpret these data and why this analysis was conducted in this way.

- For each of the analyses, it would help to get an example of one of the models, and would facilitate transparency as well. Were the models for Table 1 something like: “tempo ~ emotion * piece + (1|subject)”? For Tables 2 and 3, something like “sadness ~ tempo + pitch + … + (1|subject)”? Were random effects or intercepts used at all?

- It is not clear to me how to interpret the Pseudo R^2 values. On pg. 21 (line 421-422) the authors say: “The marginal Pseudo R^2 values in Table 2 suggest that overall, the cue values and combination models used to portray the intended emotions were highly significant”. Since significance is usually reserved to mean statistically significant, is that what is meant here? If so, how is that inference drawn? If that’s not what is intended, perhaps this needs to be reworded.

- In a few places, the authors say that the cue combinations in Experiment 1 were better than Experiment 2. I do not understand how they can make this assertion without having run additional experiments asking participants how well the “productions” represented the intended emotions.

o Pg. 43 ““When we compare how well the cue combinations represent the intended emotions, we find that overall, the cue-emotion profiles in Experiment 1 scored remarkably higher than the cue combinations used in Experiment 2.” Higher in what?

o Pg. 48 “However, the production approach created cue combinations that were better representatives of the different emotional expressions than the ones produced with the systematic manipulation approach”. How were “better” or “worse” representations defined here?

- The authors sometimes refer to the perception approach as “systematic manipulation” and sometimes as “evaluation”, which I find a bit confusing. It would be helpful to pick one of these terms and stick with it.

Reviewer #2: This submission presents the results of two experiments on music and emotion, the first in which participants could modify seven different aspects ('cues') of musical excerpts, and the second in which the cues were manipulated factorially (as levels of independent variables). These two approaches--production vs systematic manipulation, both found in the prior literature--are contrasted to one another.

The study is well-designed, executed carefully, and meticulously analyzed. For the most part the study reinforces what we already know from the many extant studies on music and emotion: tempo (faster vs slower), mode (major vs minor), articulation (notes played more connected or more separated), and dynamics (louder vs softer) all played a role. The statistics take a bit of care to read (and their introduction for Experiment 2 covers much of the same ground as for Experiment 1) but with due attention all can be understood (readers may not immediately grasp that, in the analyses, minor mode is shown with positive values and major mode with negative values, the opposite of what one would expect a priori).

To this reviewer's eye the most interesting contribution is in the area of timbre/instrumentation, which the senior author has engaged for some time. The difference in the kind of instrumental sounds used (ensembles vs the more typical solo instrument) and the simultaneous manipulation of brightness (via variable LPF) makes comparison with prior studies (e.g the explicit reference/comparison to Balkwill & Thompson) more difficult.

This reviewer also hoped for more detailed discussion of the relationships between the cues and emotional overlap/differentiation but what the paper presents is adequate.

There is some reduplication in the exposition, especially at the beginning, where the aims and setup of the study is presented more than once, and in the discussion and conclusion; this is not unusual in dissertations, from which this study may be derived. No substantial revision is necessary but some consideration could be given to tightening the introductory materials and conclusion.

6. PLOS authors have the option to publish the peer review history of their article (what does this mean?). If published, this will include your full peer review and any attached files.

Reviewer #1: No

Reviewer #2: No

---

## [Author Response · Author response to Decision Letter 0]

6 Dec 2022

Reviewer #1: I appreciated the opportunity to review “Emotional expression through musical cues: A comparison of production and evaluation approaches”. In this work, the authors directly compared the results of a perception vs. production approach to the study of cues underlying musical emotions. Overall, they found that generally the same cues were used in both perception and production of emotions. Based on this, they suggest that production may be a more efficient approach to the study of musical emotions, since multiple cues can be simultaneously incorporated.

In general, my impression of the work is positive. I especially enjoyed the discussion about how the production approach could be used to test different populations. It’s also great to see that the authors were able to implement the EmoteControl online, which opens the door to many exciting future studies.

R: Thank you. 

There are several points that I would like to see addressed before I can confidently recommend the work for publication. I have some minor comments about how their findings are interpreted (see below), which I’m confident the authors will be able to address. My more major comment about the work is that it’s not clear to me how it offers a novel contribution to the overall discussion on this topic. At the same time, my position is that novelty is overrated in the publication process, and my preference as a scholar is to have as many well-conducted studies in the literature as possible, regardless of novelty. So I would like to hear more from the authors about the specific significance of this work, but I do not think that a high degree of novelty is necessary for publication of the work. Therefore, my recommendation is for the authors to undertake minor revisions.

R: These are all very good points, and we will address the interpretation of the findings later on, and just to note we have now highlighted the significance of the work in more practical terms and downplayed the overused “novelty” card (mentions of ‘novel’ and ‘new’ have now been omitted from the manuscript - pg 7, 8, 49, 54, 55, 58).

- The main goal of the work seems to be to directly compare a perception approach vs. a production approach. As the authors acknowledge, there are still quite substantial (though necessary) differences between the two paradigms they use – different groups of participants, the number of levels available to the participants, etc. In this sense, I’m not totally sure how this approach offers anything unique above and beyond their review of the literature comparing perception and production studies that have already been conducted. Perhaps there is additional usefulness in using the same cues and the same excerpts? A clearer explanation of the benefits of this specific approach would be useful.

R: This is a fair point and deserves attention and better handling by us. First of all, we consider the production approach to be valuable for pragmatic reasons; consider the amount of cues that may contribute to emotions: in Handbook of music and emotions (2010) Gabrielsson and Lindström list 20 musical cues that have been studied in isolation in relation to emotions via perception studies over the last 90 years. We cannot exhaust the combinations of the possibilities with a traditional perception approach (systematic manipulation experiments) in a reasonable time, since even the last 90 years have mostly brought us studies using a small combination of cues (2-4) and only a limited number of studies (around 4 - Eerola, Friberg, & Bresin, 2013; Juslin, 1997; Juslin & Lindström, 2010; Scherer & Oshinsky, 1977) have looked at a bigger number of cue combinations across 40-60 experiments carried out throughout these years. To make progress in learning how cue combinations communicate emotions in real music within a reasonable time frame, we argue that a production approach offers more efficiency than a perception approach. Naturally a cautious approach would be to keep the production studies in control by occasionally carrying out studies using a factorial manipulation design. Nevertheless, the fact that a production approach creates the possibility of moving into the actual study of multiple cue combinations in an economic fashion is where the real significance of this approach is coming from.

We have now attempted to clarify the significance of the production approach and the comparison of approaches in the manuscript: pages 7, 8, 58, 59.

- I find the analyses somewhat confusing. My first confusion is that I’m not sure how to interpret the effects of Piece reported in Table 1 (or even why it’s included as a predictor in the first place). Does it mean, for example, that some pieces were just generally chosen to be faster than other pieces (presumably because compositional cues suggested different optimal tempos)? Would it make more sense to treat Piece as a random factor? I’m not suggesting it is necessary for the authors to do this; I’m simply trying to better understand how I should interpret these data and why this analysis was conducted in this way.

R: The Likelihood ratio tests for the Piece factor were made explicitly to explore whether the pieces, which varied in musical structure, may have affected how certain cues were used in the portrayal of emotions. In fact, Table 1 shows us that the Piece factor had a significant effect on tempo, articulation, dynamics, mode, and instrumentation cues, which suggests that the cues may have been used differently when portraying the same emotion across pieces. At this initial stage of the analysis, we just wanted to show that cue contributions may have differed across pieces when attempting to portray the same emotion, and therefore, we would rather not hide the musical pieces as a random factor because we think it is clearer to contain them explicitly in the analysis. However, we follow your suggestion in the next analyses (Tables 2 and 3) where Participant and Piece are considered as Random factors, since the aim of the subsequent analyses was to explore how the cues were used in combination across the pieces to portray specific emotions. 

- For each of the analyses, it would help to get an example of one of the models, and would facilitate transparency as well. Were the models for Table 1 something like: “tempo ~ emotion * [+] piece + (1|subject)”? For Tables 2 and 3, something like “sadness ~ tempo + pitch + … + (1|subject)”? Were random effects or intercepts used at all?

R: You are close, in Table 1, we first started with a base model for each of the cues in relation to the intercept (here we will give an example using the tempo cue): Tempo ~ 1 + (1|Participant), which was then updated with the factors (Emotion, Piece, Emotion x Piece) individually: Tempo ~ 1 + Emotion + (1|Participant), Tempo ~ 1 + Piece + (1|Participant), and so on. An example of the base model and an updated model with the Emotion factor have now been given in brackets in the main text on page 18 lines 389-392.

Participant and Piece factors were inputted as random factors in the models in both Tables 2 and 3. The LMM models used in Table 2 (the production study) were as follows: Cue ~ Emotion + (1|Piece) + (1|Participant). For example, Tempo ~ Sadness + (1|Piece) + (1|Participant). This has now been given in brackets on page 20 lines 419-421. In Table 3 (the perception study), LMM models were computed to explore how the cues impacted the ratings of the different emotion scales. For each target emotion, the following model was computed: Rating ~ Tempo + Pitch + Mode + Dynamics + Brightness + Articulation + Strings + Woodwinds + Brass + (1|Piece) + (1|Participant). This example of the model has been inputted in the main text in brackets on page 34 lines 678-680.

- It is not clear to me how to interpret the Pseudo R^2 values. On pg. 21 (line 421-422) the authors say: “The marginal Pseudo R^2 values in Table 2 suggest that overall, the cue values and combination models used to portray the intended emotions were highly significant”. Since significance is usually reserved to mean statistically significant, is that what is meant here? If so, how is that inference drawn? If that’s not what is intended, perhaps this needs to be reworded.

R: Pseudo R^2 values are mainly used to give an indication of the proportion of variability explained by the model. In this case, it presents the proportion of variability of the different target emotions explained by the cue combinations in the models. However, the significance of the cues is derived from the GLMM models. Since Pseudo R^2 values are explained in the sentence prior to the one in question, we have omitted the sentence in question as it did not offer further useful information (see page 26 lines 495-498). 

- In a few places, the authors say that the cue combinations in Experiment 1 were better than Experiment 2. I do not understand how they can make this assertion without having run additional experiments asking participants how well the “productions” represented the intended emotions.

R: The comparisons were being made by looking at the Pseudo R^2 values of the cue-emotion models, where the ones produced in Experiment 1 were higher than the ones created in Experiment 2, suggesting that Experiment 1’s cue-emotion models are able to better capture the variance in the values. However, you are right, the wording is ambiguous. In line with this point and the next two, we modified this assertion to be more neutral on page 52 lines 988 - 993: “When we compare the Pseudo R^2 marginal values of the cue-emotion models, we find that overall, the cue-emotion profiles in Experiment 1 scored noticeably higher than the ones used in Experiment 2. This indicates that the cue-emotion models used by participants in the production study are able to capture the variation more fully than the cue-emotion models produced in the perception study.” 

The text on page 58 lines 1151 - 1153 was also modified: “Moreover, the production approach created cue-emotion models that had a higher predictive model accuracy score than the ones produced in the perception study.”

- Pg. 43 ““When we compare how well the cue combinations represent the intended emotions, we find that overall, the cue-emotion profiles in Experiment 1 scored remarkably higher than the cue combinations used in Experiment 2.” Higher in what?

R: Good point, it was unclear that we are still talking about the Pseudo R^2 values and we have now rectified this explanation (please see previous point).

- Pg. 48 “However, the production approach created cue combinations that were better representatives of the different emotional expressions than the ones produced with the systematic manipulation approach”. How were “better” or “worse” representations defined here?

R: You are right, perhaps the word “representation” is misleading here since representation may allude to underlying representations of some kind, which is not really at stake here. We have now avoided using this word and simply refer back to cue-emotion model accuracy (Pseudo R^2): “Moreover, the production approach created cue-emotion models that had a higher predictive model accuracy score than the ones produced in the perception study.” - page 58, lines 1151 - 1153

The text on page 53 lines 1030-1033 was also modified to avoid using the term ‘represented’ and now reads as follows: “Although sadness and joy (basic emotions) had two of the highest predictive model accuracy scores in this paper, this theory does not explain the high ranking of the cue-emotion model for calmness,...”

- The authors sometimes refer to the perception approach as “systematic manipulation” and sometimes as “evaluation”, which I find a bit confusing. It would be helpful to pick one of these terms and stick with it.

R: Good point and we’ve made the terms consistent throughout the paper, and actually we would like to hijack the term the reviewer used (perception, rather than ‘systematic manipulation’ or ‘evaluation’). We will express our gratitude for these valuable suggestions in the acknowledgments.

Reviewer #2: This submission presents the results of two experiments on music and emotion, the first in which participants could modify seven different aspects ('cues') of musical excerpts, and the second in which the cues were manipulated factorially (as levels of independent variables). These two approaches--production vs systematic manipulation, both found in the prior literature--are contrasted to one another.

The study is well-designed, executed carefully, and meticulously analyzed. For the most part the study reinforces what we already know from the many extant studies on music and emotion: tempo (faster vs slower), mode (major vs minor), articulation (notes played more connected or more separated), and dynamics (louder vs softer) all played a role. The statistics take a bit of care to read (and their introduction for Experiment 2 covers much of the same ground as for Experiment 1) but with due attention all can be understood (readers may not immediately grasp that, in the analyses, minor mode is shown with positive values and major mode with negative values, the opposite of what one would expect a priori).

R: Thank you for your kind comments. We have now clarified some of the statistical analyses by providing an example of the models used, as suggested by Reviewer 1. Furthermore, we have omitted the introductory explanation of how results in Table 3 should be interpreted (on page 37), since this is identical to the LMM table (Table 2) in Experiment 1. Instead, on page 34 lines 681-685, we direct the reader to the initial explanation given for Experiment 1.

To this reviewer's eye the most interesting contribution is in the area of timbre/instrumentation, which the senior author has engaged for some time. The difference in the kind of instrumental sounds used (ensembles vs the more typical solo instrument) and the simultaneous manipulation of brightness (via variable LPF) makes comparison with prior studies (e.g the explicit reference/comparison to Balkwill & Thompson) more difficult.

R: True, the role of timbre is challenging to directly compare between the studies as many of the past studies have used the timbre in isolation and here we have two-fold changes of timbre in the text (brightness and different instruments and instrument families), which perhaps makes the interpretation challenging when we compare the current study to work by Balkwill and Thompson (e.g., bansuri flute and stringed instrument such as sitar). The use of instrument ensembles versus solo instruments, and two dimensions of timbre (brightness and also different instruments) are mentioned on page 49–50, together with a suggestion that future studies should expand research on potential connections between these two dimensions of timbre (pg 50 lines 957-960). We would suggest that the bigger picture here is to consider this study as a proof of concept that the production approach is a viable alternative to explore the cues and this would now open more opportunities to do more carefully managed timbre options/choices. 

This reviewer also hoped for more detailed discussion of the relationships between the cues and emotional overlap/differentiation but what the paper presents is adequate.

R: Fair point. Since the findings of the individual experiments (the Results sections of Experiment 1 and 2) presented the cue combinations used per emotional expression, we thought to then discuss the differentiation of cues and emotion from the cues’ perspective in the Comparison of Experiments section, to highlight how the individual cues’ usage varied across experiments. A summary of cue levels used to shape the different emotions in the two experiments is also provided in Table 4. Secondly, we deliberately didn’t engage in extensive theorising concerning the emotions and the sources of the expressive cues, that could have then explained why there are some redundancies/overlaps in the cues, but this is perhaps a good direction for future studies, which could take a theoretical stance as a starting point for a series of cue production experiments (e.g., co-production of cues related to the physiological states…).

There is some reduplication in the exposition, especially at the beginning, where the aims and setup of the study is presented more than once, and in the discussion and conclusion; this is not unusual in dissertations, from which this study may be derived. No substantial revision is necessary but some consideration could be given to tightening the introductory materials and conclusion.

R: Thanks, it is always good to tighten up the manuscript and we indeed spotted several repetitions, although some of these we have left in as signposting of what is coming next. But generally we have shortened all the introductions of the sections to avoid the duplication: overall introduction page 9 line 188, Experiment 1 intro page 10 line 219, Experiment 2 intro page 27 line 532, Experiment 2 results intro page 33 line 663, Experiment 2 statistics intro page 37 line 693, and the introductory paragraph of the Comparison section page 40 line 788.

---

## [Editor Report · Decision Letter 1]

12 Dec 2022

Emotional expression through musical cues: A comparison of production and perception approaches

PONE-D-22-22560R1

Dear Dr. Grimaud,

We’re pleased to inform you that your manuscript has been judged scientifically suitable for publication and will be formally accepted for publication once it meets all outstanding technical requirements.

Kind regards,

Sukru Torun

Academic Editor

PLOS ONE

Additional Editor Comments (optional):

Thank you for your effort to revise your manuscript, taking into account the really valuable comments from the reviewers. I congratulate and thank you in the belief that the revised version of the article reflects the content more clearly and will be more beneficial to the readers.
---

## [Editor Report · Acceptance letter]

20 Dec 2022

PONE-D-22-22560R1 

Emotional expression through musical cues: A comparison of production and perception approaches 

Dear Dr. Micallef Grimaud:

I'm pleased to inform you that your manuscript has been deemed suitable for publication in PLOS ONE. Congratulations! Your manuscript is now with our production department. 

Kind regards, 

on behalf of

Prof. Dr. Sukru Torun 

Academic Editor

PLOS ONE